# Improved Survival after Breast-Conserving Therapy Compared with Mastectomy in Stage I-IIA Breast Cancer

**DOI:** 10.3390/cancers13164044

**Published:** 2021-08-11

**Authors:** Ivica Ratosa, Gaber Plavc, Nina Pislar, Tina Zagar, Andraz Perhavec, Pierfrancesco Franco

**Affiliations:** 1Department of Radiation Oncology, Institute of Oncology Ljubljana, 1000 Ljubljana, Slovenia; iratosa@onko-i.si (I.R.); gplavc@onko-i.si (G.P.); 2Faculty of Medicine, University of Ljubljana, 1000 Ljubljana, Slovenia; aperhavec@onko-i.si; 3Department of Surgery, Institute of Oncology Ljubljana, 1000 Ljubljana, Slovenia; npislar@onko-i.si; 4Department of Epidemiology and Cancer Registry, Institute of Oncology Ljubljana, 1000 Ljubljana, Slovenia; tzagar@onko-i.si; 5Department of Translational Medicine, University of Eastern Piedmont, 28100 Novara, Italy; 6Radiation Oncology Unit, AOU “Maggiore della Carità”, 28100 Novara, Italy

**Keywords:** early-stage breast cancer, breast-conserving therapy, mastectomy, radiation therapy, outcome

## Abstract

**Simple Summary:**

The majority of patients with breast cancer are suitable for either breast-conserving therapy, consisting of breast-conserving surgery and radiation therapy, or mastectomy alone. In the present study, we compared survival outcomes in 1360 patients affected with early-stage breast cancer (stage I-IIA) according to the type of local treatment. We confirmed that patients treated with breast-conserving therapy had a lower rate of local, regional, and distant disease recurrences, and at least equivalent overall survival compared to those treated with mastectomy alone. Our results add to previous research showing a potential benefit of breast-conserving therapy when compared to mastectomy in patients suitable for both treatments at baseline.

**Abstract:**

In the current study, we sought to compare survival outcomes after breast-conserving therapy (BCT) or mastectomy alone in patients with stage I-IIA breast cancer, whose tumors are typically suitable for both locoregional treatments. The study cohort consisted of 1360 patients with stage I-IIA (T1–2N0 or T0–1N1) breast cancer diagnosed between 2001 and 2013 and treated with either BCT (*n* = 1021, 75.1%) or mastectomy alone (*n* = 339, 24.9%). Median follow-ups for disease-free survival (DFS) and overall survival (OS) were 6.9 years (range, 0.3–15.9) and 7.5 years (range, 0.2–25.9), respectively. Fifteen (1.1%), 14 (1.0%) and 48 (3.5%) patients experienced local, regional, and distant relapse, respectively. For the whole cohort of patients, the estimated 5-year DFS and OS were 96% and 97%, respectively. After stratification based on the type of local treatment, the estimated 5-year DFS for BCT was 97%, while it was 91% (*p* < 0.001) for mastectomy-only treatment. Inverse probability of treatment weighting matching based on confounding confirmed that mastectomy was associated with worse DFS (HR 2.839, 95% CI 1.760–4.579, *p* < 0.0001), but not with OS (HR 1.455, 95% CI 0.844–2.511, *p* = 0.177). In our study, BCT was shown to have improved disease-specific outcomes compared to mastectomy alone, emphasizing the important role of adjuvant treatments, including postoperative radiation therapy, in patients with early-stage breast cancer at diagnosis.

## 1. Introduction

Long-term follow-up data from several randomized controlled trials undertaken to compare the efficacy of mastectomy with that of breast-conserving therapy (BCT) consisting of lumpectomy or quadrantectomy followed by postoperative radiotherapy showed no differences in terms of disease-free survival (DFS), distant-disease-free survival, and overall survival (OS) among the treatment groups [1,2,3,4,5]. The trials, conducted in the 1970s and 1980s, paved the way for the increased utilization of BCT for patients with stage I-II breast cancer [6]. Retrospective studies based on the analysis of large patient populations with early-stage breast cancer (EBC) treated in modern contexts indicate that BCT is at least comparable or even better in terms of breast-cancer-specific survival (BCSS) and OS compared to mastectomy without radiation therapy for early-stage breast cancer [7,8,9,10,11,12,13]. Improved DFS and OS rates were recently confirmed in smaller cohort studies [14,15,16,17,18,19,20] and were also observed in young women with EBC [21,22,23]. Based on older research, local recurrence is considered to be more common after BCT than after mastectomy. However, over the last 20 years, local recurrence rates after BCT have decreased substantially and are now shown to be as low as after mastectomy, most likely due to better understanding of breast cancer heterogeneity and consequently tailored systemic and radiation treatments [24,25].

For patients with breast cancer who are potentially suitable for both local treatments, BCT is suggested as the evidence-based primary local treatment [26]. Breast-conserving surgery is one of the major advances in the management of breast cancer. Nevertheless, breast-conserving surgery rates, as commonly reported, have reached a plateau in the last decade. In 2010–2011, breast-conserving surgery rates for the general population with unilateral EBC were 64.5–69% in the US [27,28,29], 73.3% in Europe [30], and 36.5–49.4% for patients younger than 50 years [28,31]. However, in some countries, the overall rate of breast-conserving surgery is lower than 50% [32,33].

The aim of BCT, as compared to mastectomy, is to achieve oncological safety with less extensive surgery, minimizing the psychosocial sequelae, and to obtain favorable cosmetic results with the use of oncoplastic techniques whenever possible. Most women feel negatively impacted by the scars resulting from breast cancer surgery, a consequence that particularly affects patients undergoing mastectomy [34]. When comparing long-term quality of life amongst patients treated with BCT or mastectomy, significantly lower scores for body image, role, and physical and sexual functioning, and more lifestyle disruptions were found for those treated with mastectomy. Conversely, psychosocial functioning slowly increased over time for women who underwent BCT, regardless of their age [35,36]. For patients undergoing mastectomy, access to breast reconstruction procedures is essential to increase positive body image and overall satisfaction and to maintain health-related quality of life [26,37]. However, barriers to immediate or delayed breast reconstruction exist on many levels, depending on the type of hospital (teaching vs. private), geographic location, reimbursement by insurance companies, lack of patient awareness, and the acceptability of the procedures by both physicians and patients [38]. In the past decade, the use of breast reconstruction has increased globally together with the indications for the use of postmastectomy irradiation for EBC [39,40]. It has been documented that the use of postmastectomy radiation therapy may be a negative outcome predictor for breast reconstruction [39]. On the other hand, opting for breast reconstruction may also influence clinical decision making in recommending postoperative radiation therapy due to the impact of radiation on long-term cosmetic results [39,41].

In the current study, we aimed to compare the outcomes of BCT and mastectomy-only treatment performed in the modern era in patients with stage I-IIA breast cancer, whose tumors are typically suitable for either BCT or mastectomy. The primary outcomes were DFS and OS, and secondary outcomes included any breast cancer recurrence (local, regional, and/or distant).

## 2. Materials and Methods

### 2.1. Study Cohort and Data Collection

For the present study, data were retrospectively collected and retrieved. The study cohort consisted of patients with stage I-IIA (T1–2N0 or T0–1N1) breast cancer diagnosed between 2001 and 2013 and treated with upfront BCT or mastectomy only with or without reconstruction, achieving clear resection margins in both cases. General patient demographic, histological characteristic, systemic therapy (chemotherapy, endocrine, or anti-HER2 therapies), and local treatment (surgery, radiotherapy) data were collected from individual clinical records. Clinical follow-up information for all patients was updated until 31 March 2021. For the purpose of this study, patient disease stage was classified based on medical records, according to breast carcinoma *TNM Classification of Malignant Tumours* (7th edition) [42]. Intrinsic subtypes of BC were defined as luminal A-like (estrogen-receptor positive, ER+; human epidermal growth factor receptor 2 negative, HER2−; Ki67 < 20%, progesterone receptor positive (PR+) with a cut-point of ≥20%); luminal B-like HER2-negative (ER+, HER2−, Ki67 ≥ 20%, or low PR+); luminal B-like HER2-positive (ER+, HER2+, any PR, any Ki67); HER2-positive (HER2+, PR−, ER−); and ‘Basal-like’ (ER−, PR−, HER2−) according to clinicopathological surrogate definitions as defined in the St Gallen International Expert Consensus on the primary therapy of EBC in 2013 [43].

### 2.2. Statistical Analysis

To compare clinical and tumor characteristics between the two groups (patients treated with BCT vs. those treated with mastectomy) we used independent-sample t-tests for continuous variables and Pearson’s χ^2^ or Fisher’s exact tests for categorical variables. Data were expressed as median with a sample range for continuous variables, and as counts with frequencies for categorical data. The Kaplan–Meier method was used to calculate estimated survival curves, and the log-rank test was used to compare the two groups. Univariate and multivariate Cox’s proportional hazards models were used to assess the effects of covariates on survival. The effect sizes were given as hazard ratios (HR) with 95% confidence intervals (CI). Overall survival (OS) was specified as the time of BC diagnosis to the date of death or last follow-up. Disease-free survival (DFS) was defined as time to disease recurrence (any type; local and/or regional and/or distant) after surgery. All tests were two-sided, and the statistical level of significance was set to *p* < 0.05. In addition, to reduce bias in our observational study and to analyze the cohort with a representative distribution of matching factors, we used a propensity-score-matched analysis with an inverse probability of treatment weighting (IPTW), calculating the reciprocal of the probability of receiving the treatment that the patient in fact received. We performed adjusted analyses using a multiple linear regression model and IPTW using the propensity score estimated via logistic regression. The outcome model used in the IPTW analysis was a linear regression of outcome on BCT or mastectomy, weighted by the estimated propensity score [44,45]. Covariate adjustment was based on age at diagnosis, pathological tumor and nodal stage, overall breast cancer stage group, tumor grade, type of axillary surgery, and the receipt of chemotherapy and/or endocrine therapy. Data were analyzed using IBM SPSS Statistics software version 26 (Statistical package for the Social Sciences Statistical Software, SPSS Inc, IBM Corporation, Armonk, NY, USA).

## 3. Results

In total, 1360 patients with stage I-IIA breast cancer treated with either BCT (*n* = 1021, 75.1%) or mastectomy only (*n* = 339, 24.9%) were included in the study. The median age at breast cancer diagnosis was 61 years in both groups (BCT range 23–87 and mastectomy range 27–91). The majority of breast tumors were sized ≤2 cm (*n* = 1100, 81.5%) and had positive estrogen or progesterone receptors (*n* = 1241, 92.3%). More than half of the patients had left-sided breast cancer (*n* = 729; 54.2%). Compared with patients receiving mastectomy only as local therapy, patients undergoing BCT were less likely to be younger than 50 or older than 70 years, less likely to have had more extensive axillary surgery, and more likely to have stage I breast cancer. We observed no differences in the two groups with respect to tumor grade, intrinsic subtype, and administration of endocrine or targeted therapy. However, more patients in the mastectomy group received adjuvant chemotherapy (Table 1).

### 3.1. Surgery

Most patients received a limited axillary surgery (*n* = 1360, 81.3%). Data regarding breast reconstruction were known for 557 (41.0%) patients. Among the 62 patients receiving breast reconstruction, 4 had stage IA, 31 had stage IB, and 27 had stage IIA breast cancer.

### 3.2. Radiation Therapy

Out of the 1021 patients receiving radiotherapy, all following breast-conserving surgery, 495 (48.5%) were treated with conventional fractionation (median dose 50 Gy; range 28–50.4 Gy), 489 (47.9%) with moderate hypofractionated schedules (median dose 45 Gy; range 34.5–47.5), and 37 (3.6%) received one of the ultra-hypofractionated schedules (median dose 31.5 Gy; range 20–50 Gy). Additional dose to the tumor bed was received by 661 (64.7%) patients. Almost all patients received whole-breast radiotherapy only, excluding axillary or supraclavicular nodal volumes (*n* = 1014; 99.3%).

### 3.3. Systemic Treatment

Timing (preoperative versus postoperative) and type of systemic treatment are presented separately for both groups in Table 2. The receipt of taxane-based chemotherapy was more frequently observed in the mastectomy group, as compared to the BCT group (32.8% versus 24.7%; *p* < 0.0005).

In total, 578 (45.1%) evaluated patients received endocrine therapy with aromatase inhibitors, 539 (42.0%) with tamoxifen, and 122 (12.9%) with treatment combinations (i.e., tamoxifen and aromatase inhibitors). Compared with patients treated with mastectomy only, patients undergoing BCT more often received adjuvant endocrine therapy with aromatase inhibitors (*n* = 141; 40.3% versus *n* = 437; 46.9%) and were less likely to receive adjuvant therapy with tamoxifen (152; 43.4% versus 387; 41.5%). The observed differences were statistically significant (*p* < 0.0005).

### 3.4. Outcome

Median follow-up for DFS and OS was 6.9 years (range: 0.3–15.9) and 7.5 years (range: 0.2–25.9), respectively. Overall, 86 (6.3%) patients experienced local (LR), regional (RR), or distant recurrence. We observed statistically meaningful differences across all recurrence types (Table 3). The cumulative incidences of 5-year LR and 10-year LR were 2.0% and 3.0% for the whole group, respectively. For the BCT group, the cumulative incidences of 5-year and 10-year LR were 1.0% and 3.0%; the corresponding cumulative incidences for the mastectomy group were 4% for both observed intervals. Observed absolute differences in cumulative incidence of recurrences between BCT and mastectomy groups were small, although statistically significant.

For the whole cohort, the estimated 5-year and 10-year DFS were 96% and 95%, respectively. After stratification according to the type of local treatment, the estimated 5-year and 10-year DFS for the BCT group were 97% and 96%, and the estimated 5-year and 10-year DFS for the mastectomy-only group were 91% and 90% (log-rank; *p* < 0.001). In a univariate analysis, the following factors were associated with a decreased DFS: not receiving BCT, age > 70 years, tumor size ≥ 2 cm, stage IIA, tumor grade ≥ 2, and omission of endocrine adjuvant therapy. Upon multivariate Cox analysis, type of local treatment and tumor grade maintained statistical significance (Table 4).

The estimated 5-year and 10-year OS, calculated for the whole cohort, were 97% and 92%, respectively. Among patients treated with BCT, 5-year and 10-year OS estimates were 97% and 93%, and for those treated with mastectomy-only, 95% and 89%, respectively (log-rank; *p* = 0.045). Univariate and multivariate analyses were performed to analyze factors associated with worse survival outcomes. The use of mastectomy as the only local treatment was found to be correlated with worse OS in univariate analysis, but this finding was not confirmed in the multivariate Cox regression analysis (Table 5).

High tumor grade and the omission of systemic chemotherapy were the only two factors associated with poorer OS. The type of local treatment was not associated with improved OS using multivariable Cox regression. Kaplan–Meier survival curves for all included patients representing DFS and OS with respect to breast cancer stage group are presented in Figure 1.

After IPTW matching based on confounding variables (age at diagnosis, pathological tumor and nodal stage, overall breast cancer stage group, tumor grade, type of axillary surgery, and the receipt of chemotherapy or endocrine therapy), and after excluding study subjects with extreme weights, 1343 patients were available for the analysis. Mastectomy was associated with worse DFS (HR 2.839, 95% CI 1.760–4.579, *p* < 0.0001). This finding was not confirmed for OS (HR 1.455, 95% CI 0.844–2.511, *p* = 0.177).

## 4. Discussion

In our patient cohort treated in the contemporary era, we evaluated the therapeutic outcomes in patients with stage I-IIA breast cancer treated with either conservative surgery combined with postoperative radiation therapy or with mastectomy alone. We demonstrated an outcome improvement associated with BCT in terms of both 5-year DFS and 5-year OS, with absolute differences of 6% and 2%, respectively. The observed differences were also consistent at 10 years. However, using multivariable Cox regression and after IPTW matching, the only outcome that was found to be significantly impacted by the type of local treatment (BCT compared to mastectomy) was DFS.

In our study, we observed very low LR, RR, and distant metastasis rates within the whole cohort. The cumulative in-breast/chest-wall failure rates at 10 years after treatment were 3.0% for the BCT group and 4.0% for the mastectomy group. Compared to historical series, reporting approximately 5–10% in-breast failures after 10 years from treatment, our results compare favorably [46,47]. This finding is in agreement with the reports of other studies, which are all convincing and uniformly show DFS and/or OS benefits for BCT compared to mastectomy in patients with stage I-II breast cancer (Table 6). Equivalent or improved outcomes with BCT as compared with mastectomy in terms of locoregional control, BCSS, DFS, or OS have been reported regardless of age [14,15], intrinsic breast cancer subtype [48,49,50,51], pathological tumor stage [11,14,15,51], overall breast cancer stage [52], and grade [51]. Recently, a Swedish cohort study using prospectively collected data of women with stage T1-2 N0-2 breast cancer confirmed better survival with BCT vs. mastectomy (irrespective of radiation therapy) even when taking into account comorbidity and socioeconomic status in both node-negative and node-positive disease, pointing out that offering more extensive surgery to patients who are suitable for either breast-conserving surgery or mastectomy is not saving lives [51].

Improved survival rates in patients with breast cancer over the past couple of decades are largely attributable to the use of breast cancer screening and better imaging, predictive biomarkers, and better understanding of breast cancer heterogeneity. New advances in the field of systemic treatments, including molecularly targeted therapies, endocrine therapy, taxane-based chemotherapy, and bisphosphonates have all effectively contributed to reduce the risk of distant and local breast cancer recurrence. Consequently, breast cancer mortality rates have declined in recent decades even in patients with a low risk of recurrence at baseline [53]. However, the contributions of a particular type of local treatment and new developments related to both surgical techniques and modern radiation therapy are frequently overlooked [54]. It is well known that prevention of locoregional recurrence reduces the risk of breast-cancer-specific death and is related to improvements in OS [55,56]. Many possible reasons for the better outcomes observed for patients with breast cancer undergoing lumpectomy combined with postoperative radiotherapy compared with mastectomy have been already elucidated [46,54,57,58]. Surgery does provide superior local control within resected tissue; however, with tangential radiotherapy techniques, the treated volume is larger (as compared to simple mastectomy) and typically includes unoperated breast tissue in its entirety, part of the muscle, regional lymphatics, draining lymphatics towards the axillary region, subcutaneous lymphatic plexus, and skin [46]. Incidental irradiation may potentially sterilize microscopic disease outside the breast tissue as it covers approximately 85% of axillary level I lymph nodes if the patient is treated in a supine position [59]. At the same time, radiotherapy techniques have improved in the past years, with three-dimensional treatment planning and heart-sparing techniques, including deep-inspiration breath-hold, prone-positioning, and partial breast irradiation substantially decreasing the dose to the heart and subsequently reducing mortality rate from the cardiac events [60].

The possibility of an anticancer immune response that can be elicited by radiotherapy outside of the radiation field, targeting micro- or macro-metastases, is also one of the postulated mechanisms of action. Abscopal effects have been demonstrated not only at higher doses per fraction but also at the lower dose per fraction (2.0–2.5 Gy) typically used in breast cancer postoperative radiotherapy [61]. Moreover, it is well known that postoperative radiotherapy not only reduces local recurrence, but also diminishes the risk of any recurrence type, including distant relapse, which could be explained by mechanisms such as the abscopal effect [62].

Omitting postoperative radiation therapy after mastectomy in patients undergoing breast reconstruction or in those with N1 disease may be a partial culprit of higher disease recurrence in these patients. In our study, 17.6% of patients had pathological N1 disease and the percentage of patients with pathological N1 disease did not differ between the groups. None of the patients with pathological N1 disease in the mastectomy group received postoperative radiation therapy. In a study by Sun et al., the authors analyzed the treatment outcomes of 4262 patients with clinical stage T1-2N1M0 breast cancer, and 832 (21.6%) of them received mastectomy and postoperative radiation therapy. In multivariate and propensity-score matching analyses, radiation therapy, but not type of surgical treatment, appeared to be an independent prognostic factor for improved OS, DFS, and loco-regional recurrence [19]. Although results from the meta-analysis of individual patient data clearly demonstrated a benefit in reducing both recurrence and breast cancer mortality in women with one to three positive lymph nodes, postmastectomy radiotherapy is less often routinely recommended than radiotherapy following breast-conserving surgery in patients with pathological N1 disease [46,62]. Patients who require post-mastectomy radiation therapy following immediate breast reconstruction are exposed to a detectably higher risk of post-reconstruction complications, including infection, implant removal, and capsular contracture in patients receiving implant-based reconstruction, and fat necrosis in those receiving autologous-tissue-based reconstruction [63]. Factors influencing the final cosmetic outcome and patients’ preferences may all affect the final decision to undergo postoperative chest-wall radiation therapy.

Despite the many benefits observed with BCT, the rates of unilateral or bilateral mastectomies for patients with unilateral breast cancer, who are candidates for BCT, are on the rise [29,64]. Perceived risks such as a fear of developing a second breast cancer, a historical belief that mastectomy is a safer option, possible avoidance of long-term breast cancer surveillance imaging, and the inconvenience of daily radiotherapy treatments may all have contributed to increasing mastectomy rates, which is of concern [53,65,66,67]. Nevertheless, it is important to note that different mastectomy and reconstruction techniques may lead to various amounts of remaining breast glandular tissue, potentially increasing the risk of breast cancer residual disease or recurrence [68].

As pointed out by Dodwell et al., women with screen-detected breast cancer are more likely to undergo BCT and mammographic screening confers a survival advantage compared to symptomatic presentation [69]. In our study, in which two-thirds of patients were aged between 50 and 70 years (*n* = 880, 64.7%) and a similar proportion had stage I disease (*n* = 891, 65.5%), individual data regarding screening status were missing.

The strengths of our study include a large patient cohort and thorough analysis with adjustments for confounding factors. Nevertheless, we acknowledge the limitations of our study and the difficulties in bias elimination in observational research. Our results may have been influenced by the retrospective nature of the study, as the quality of our data depend on reliable data collection. Additionally, follow-up was short and some of the data were not available at the time of the analysis. In our study, the impact of specific patient and tumor characteristics (e.g., age, tumor grade or size, lymph node status, and the receipt of endocrine therapy and systemic therapy) were adjusted with the use of a rigorous approach to reduce the effects of confounding in the estimation of the type of local treatment effect. However, we acknowledge the limitations of the propensity score analysis and IPTW-matching method, especially that there was no adjustment for the impact of all baseline characteristics [45]. In addition, a comparison of the results of patients with stage I-IIA EBC and treated with mastectomy followed by postoperative radiation therapy would certainly add value to our research.

## 5. Conclusions

Breast-conserving surgery followed by postoperative radiotherapy was shown in our study to have superior outcomes, including local, regional, and distant recurrence of the disease, and at least equivalent OS compared to mastectomy alone in patients with early-stage I-IIA breast cancer. The observed differences were statistically significant, but the clinical differences were relatively small. Nevertheless, it is essential that patients with EBC who are suitable for either BCT or mastectomy are well informed throughout the shared decision-making process about each locoregional treatment option and the corresponding long-term outcomes.

## Figures and Tables

**Figure 1 cancers-13-04044-f001:**
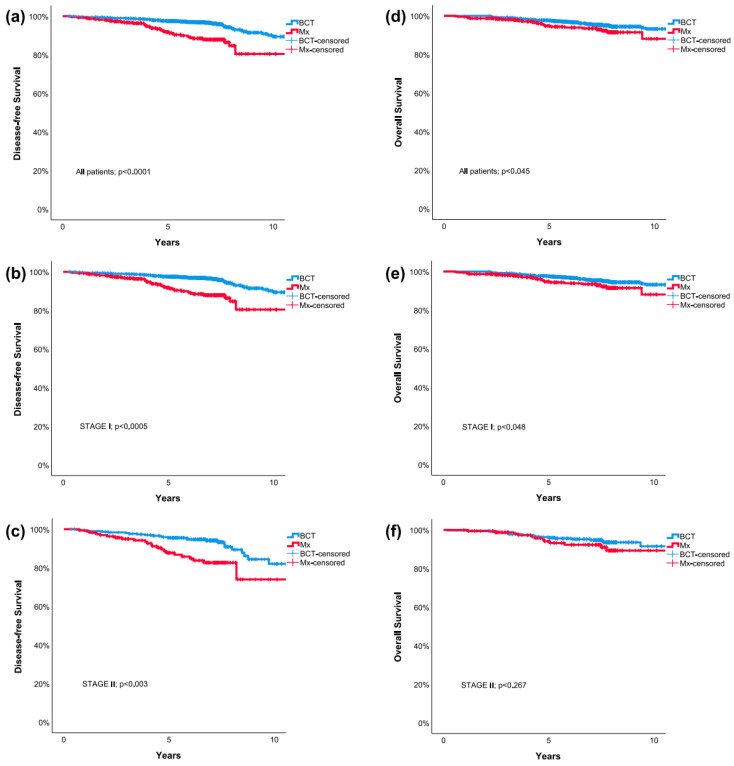
Kaplan–Meier survival curves comparing breast conserving therapy and mastectomy: Disease-Free Survival for all included patients (**a**), patients with stage I (**b**) and stage II (**c**) breast cancer; Overall Survival for all included patients (**d**), patients with stage I (**e**) and stage II (**f**) breast cancer. BCT = breast-conserving therapy; Mx = mastectomy.

**Table 1 cancers-13-04044-t001:** Patients’ clinical, pathological, and treatment characteristics.

Variable	All Patients	BCT	Mastectomy	*p* Value
	*n* = 1360 (100.0%)	*n* = 1021 (75.1%)	*n* = 339 (24.9%)	
**Age at diagnosis**				<0.0005
<50 years	246 (18.1)	159 (15.6)	87 (25.7)	
50–70 years	880 (64.7)	710 (69.5)	170 (50.1)	
>70 years	234 (17.2)	152 (14.9)	82 (24.2)	
**Year of treatment**				<0.0005
2001–2007	243 (17.9)	212 (20.8)	31 (9.10)	
2008–2013	1117 (82.1)	809 (79.2)	308 (90.9)	
**Pathological tumor stage**				<0.0005
T1	1100 (81.5)	873 (86.0)	227 (67.8)	
T2	240 (17.8)	139 (13.7)	101 (30.1)	
≤T2, details unavailable	10 (0.7)	3 (0.3)	7 (2.1)	
**Pathological node stage**				0.324
Nx	18 (1.3)	13 (1.3)	5 (1.5)	
N0	1085 (79.8)	822 (80.5)	263 (77.6)	
N0	1043 (76.7)	793 (77.7)	250 (73.7)	
N0 (ITC)	42 (3.1)	29 (2.8)	13 (3.8)	
N1	240 (17.6)	177 (17.3)	63 (18.6)	
N1 (NOS)	140 (10.3)	101 (9.9)	39 (11.5)	
N1mic	26 (1.9)	21 (2.1)	5 (1.5)	
N1a	74 (5.4)	55 (5.4)	19 (5.6)	
≤TN1, details unavailable	17 (1.3)	9 (0.9)	8 (2.4)	
**Overall stage group**				<0.0005
Stage IA	560 (41.2)	464 (45.4)	96 (28.3)	
Stage IB	331 (24.3)	255 (25.0)	76 (22.4)	
Stage IIA	469 (34.5)	302 (29.6)	167 (49.3)	
**Grade**				0.143
G1	443 (32.6)	340 (33.3)	103 (30.4)	
G2	652 (47.9)	496 (48.6)	156 (46.0)	
G3	264 (19.4)	184 (18.0)	80 (23.6)	
Unknown	1 (0.1)	1 (0.1)	0	
**Molecular subtype**				0.065
Luminal A	864 (63.5)	667 (65.3)	197 (58.1)	
Luminal B HER2−	390 (28.7)	279 (27.3)	111 (32.7)	
Luminal B HER2+	55 (4.0)	35 (3.4)	20 (5.9)	
HR−/HER2+	13 (1.0)	10 (1.0)	3 (0.9)	
HR−/HER2−	38 (2.8)	30 (2.9)	8 (2.4)	
**Axillary surgery**				<0.0005
Omitted	39 (2.9)	28 (2.7)	11 (3.2)	
SNB only	1067 (78.5)	839 (82.2)	228 (67.3)	
Axillary dissection	254 (18.7)	154 (15.1)	100 (29.5)	
**Endocrine therapy**				0.947
Yes	1239 (91.1)	929 (91.0)	310 (91.4)	
No	116 (8.5)	88 (8.6)	28 (8.3)	
Unknown	5 (0.4)	4 (0.4)	1 (0.3)	
**Chemotherapy**				0.001
Yes	411 (30.2)	271 (26.5)	140 (41,7)	
No	947 (69.7)	748 (73.3)	199 (58.3)	
Unknown	2 (0.1)	2 (0.2)	0	
**Anti-HER2 therapy**				0.256
Yes	55 (4.0)	37 (3.6)	18 (5.3)	
No	513 (37.7)	384 (37.6)	129 (38.1)	
Unknown	792 (58.3)	600 (58.8)	192 (56.6)	

Abbreviations: *n* = number; BCT = breast-conserving therapy; T = tumor; N = node; G = grade; HR = hormone receptor; HER2 = human epidermal growth receptor 2; NOS = not otherwise specified; mic = micrometastases; ITC = isolated tumor cells; SNB = sentinel node biopsy. Type of variable in bold.

**Table 2 cancers-13-04044-t002:** Type and sequence of systemic treatment.

Variable	All Patients	BCT	Mastectomy	*p* Value
	*n* = 411 (100%)	*n* = 271 (65.9%)	*n* = 140 (34.1%)	
**Chemotherapy**				0.016
Neoadjuvant systemic therapy	17 (4.1)	7 (2.6)	10 (7.1)	
Adjuvant systemic therapy	383 (93.2)	253 (93.4)	130 (92.9)	
Unknown sequence	11 (2.7)	11 (4.0)	0	
**Type of systemic chemotherapy**				<0.0005
Anthracyclines/Cyclophosphamide	236 (57.4)	172 (63.5)	64 (45.7)	
Taxanes/Cyclophosphamide	33 (8.0)	21 (7.7)	12 (8.6)	
Anthracyclines/Taxanes	80 (19.5)	46 (17.0)	34 (24.3)	
CMF	27 (6.6)	8 (3.0)	19 (13.6)	
Other drugs and combinations	13 (3.2)	9 (3.3)	4 (2.9)	
Unknown	22 (5.4)	15 (5.5)	7 (5.0)	

**Table 3 cancers-13-04044-t003:** Cumulative incidence of local, regional, and distant recurrence for patients treated with breast-conserving therapy or with mastectomy.

Site of Recurrence	All Patients	BCT	Mastectomy	*p* Value
	*n* = 1360 (100%)	*n* = 1021 (75.1%)	*n* = 339 (24.9%)	
**No recurrence**	1275 (93.8)	969 (94.9)	306 (90.3)	<0.0001
**Any recurrence**	85 (6.2)	52 (5.1)	33 (9.7)	
Local recurrence	15 (1.1)	10 (1.0)	5 (1.5)	0.001
Regional nodes or adjacent tissues/organs	14 (1.0)	8 (0.8)	6 (1.8)	0.002
Distant metastasis	48 (3.5)	34 (3.3)	14 (4.1)	<0.0001
Unknown recurrence type	8 (0.6)	0	8 (2.4)	0.018
**Second primary cancer**	47 (3.5)	29 (2.8)	18 (5.3)	0.039

Abbreviations: *n* = number; BCT = breast-conserving therapy. Type of variable in bold.

**Table 4 cancers-13-04044-t004:** Univariate and multivariate Cox regression analysis for 5-year disease-free survival.

		Univariate Analysis		Multivariate Analysis	
Variable	5-Year DFS	HR (95% CI)	*p* Value	HR (95% CI)	*p* Value
**Local treatment**			<0.0005		0.006
BCT	97%	1		1	
Mastectomy	91%	2.883 (1.868–4.449)		2.291 (1.106–4.771)	
**Age at diagnosis**			0.032		0.099
<50 years	95%	1		1	
50–70 years	97%	0.932 (0.491–1.768)		0.863 (0.479–1.556)	
>70 years	93%	1.878 (0.911–3.873)		1.424 (0.708–2.864)	
**Pathological tumor stage**			<0.0005		0.679
T1	97%	1		1	
T2	90%	3.537 (2.279–5.489)		1.628 (0.769–3.444)	
≤T2, details unavailable	88%	3.759 (0.909–15.551)		1.823 (0.411–8.088)	
**Pathological node stage**			0.267		0.354
N0	96%	1		1	
N1	96%	1.105 (0.648–1.886)		0.075 (0.003–1.701)	
N status unknown	90%	2.584 (0.936–7.138)		5.103 (0.083–313.368)	
**Overall stage group**			<0.0005		0.269
Stage IA	98%	1		1	
Stage IB	98%	0.714 (0.330–1.544)		0.558 (0.251–1.241)	
Stage IIA	93%	2.545 (1.588–4.081)		1.479 (0.653–3.346)	
**Grade**			<0.0005		<0.0005
G1	98%	1		1	
G2	97%	1.168 (0.676–2.019)		1.164 (0.651–2.082)	
G3	91%	3.144 (1.786–5.536)		2.284 (1.124–4.641)	
**Axillary surgery**			0.093		0.823
SNB only	97%	1		1	
Axillary dissection	95%	1.370 (0.840–2.235)		1.043 (0.540–2.012)	
Omitted	89%	2.564 (1.103–5.960)		3.494 (1.442–8.463)	
**Endocrine therapy**			<0.0005		0.25
Yes	97%	1		1	
No	87%	3.430 (2.059–5.715)		2.974 (1.648–5.368)	
**Chemotherapy**			0.064		0.143
Yes	93%	1		1	
No	97%	0.600 (0.389–0.924)		1.445 (0.781–2.674)	
**Anti-HER2 therapy**			0.482		0.21
Yes	96%	1		1	
No	95%	1.614 (0.386–6.752)		2.991 (0.673–13.302)	

Abbreviations: HR = hazard ratio; CI = confidence interval; BCT = breast-conserving therapy; T = tumor; N = node; G = grade; HR = hormone receptor; HER2 = human epidermal growth receptor 2; SNB = sentinel node biopsy. Type of variable in bold.

**Table 5 cancers-13-04044-t005:** Univariate and multivariate Cox regression analysis for 5-year overall survival.

		Univariate Analysis		Multivariate Analysis	
Variable	5-Year OS	HR (95% CI)	*p* Value	HR (95% CI)	*p* Value
**Local treatment**			0.047		0.689
BCT	97%	1		1	
Mastectomy	95%	1.660 (1.006–2.738)		1.060 (0.564–1.994)	
**Age at diagnosis**			0.066		0.691
<50 years	96%	1		1	
50–70 years	98%	0.932 (0.491–1.768)		0.778 (0.365–1.657)	
>70 years	93%	1.878 (0.911–3.873)		1.151 (0.469–2.824)	
**Pathological tumor stage**			0.037		0.573
T1	97%	1		1	
T2	94%	2.013 (1.212–3.344)		0.488 (0.020–11.894)	
≤T2, details unavailable	94%	1.510 (0.208–10.950)		0.701 (0.006–81.144)	
**Pathological node stage**			0.705		0.573
N0	97%	1		1	
N1	97%	0.777 (0.409–1.477)		0.271 (0.009–7.817)	
N status unknown	94%	1.169 (0.285–4.789)		0.724 (0.012–42.111)	
**Overall stage group**			0.001		0.804
Stage IA	99%	1		1	
Stage IB	95%	2.966 (1.576–5.582)		0.186 (0.005–6.430)	
Stage IIA	95%	2.338 (1.270–4.304)		0.567 (0.116–2.774)	
**Grade**			0.029		0.022
G1	99%	1		1	
G2	97%	1.780 (0.980–3.232)		1.454 (0.623–3.395)	
G3	95%	1.982 (0.992–3.959)		1.310 (0.420–4.086)	
**Axillary surgery**			0.43		0.686
SNB only	97%	1		1	
Axillary dissection	97%	0.688 (0.361–1.311)		1.204 (0.441–3.286)	
Omitted	92%	1.296 (0.406–4.139)		0.784 (0.013–47.085)	
**Endocrine therapy**			0.665		0.582
Yes	97%	1		1	
No	95%	1.243 (0.596–2.589)		0.928 (0.321–2.682)	
**Chemotherapy**			0.554		0.042
Yes	97%	1		1	
No	97%	1.294 (0.760–2.202)		1.858 (0.744–4.638)	
**Anti-HER2 therapy**			0.931		0.452
Yes	91%	1		1	
No	95%	0.960 (0.382–2.413)		0.667 (0.239–1.864)	

Abbreviations: HR = hazard ratio; CI = confidence interval; BCT = breast-conserving therapy; T = Tumor; N = node; G = grade; HR = hormone receptor; HER2 = human epidermal growth receptor 2; SNB = sentinel node biopsy. Type of variable in bold.

**Table 6 cancers-13-04044-t006:** Published studies comparing breast-conserving therapy with mastectomy for patients with stage I-II breast cancer.

Study	Years of Treatment	Stage	Number of Included Patients	Measured Outcome	BCT	Mastectomy	*p* Value
Hartman-Johnsen et al., 2015 [14]	1998–2008	T1–2	BCT: 8065	5-year BCSS	97%	88%	NA
		N0–1	Mastectomy: 4950				
Hartman-Johnsen et al., 2017 [15]	1998–2009	T1–2	All: 6387	10-year BCSS	97% (T1N0)	96% (T1N0)	NA
		N0–1			98% (T1N1)	89% (T1N1)	
Wang et al., 2018 [16]	1999–2014	T1	BCT: 1296	5-year DFS	95.30%	90.20%	0.001
		N0–1mi	Mastectomy: 4841	5-year DMFS	97%	92.20%	<0.001
				5-year OS	99.10%	96.10%	0.001
de Boniface et al., [20]	2000–2004	T1–2	BCT: 2338	13-year OS	79.50%	64.30%	<0.001
		N0–2 *	Mastectomy: 429	13-year BCSS	90.50%	84%	<0.001
Corradini et al., 2019 [17]	1998–2014	T1–2	BCT: 6412	10-year DMFS	89.40%	85.50%	0.013
		N0–1	Mastectomy: 1153	10-year OS	85.30%	79.30%	0.011
Lan et al., 2019 [18]	1998–2016	T1–2	BCT: 410	5-year LRFS	89%	85.30%	0.012
		N0–1	Mastectomy: 1406				
Sun et al., 2020 [19]	2009–2014	T1–2	BCT: 404	5-year DFS	96.50%	92.70%	0.001
		N0–1	Mastectomy (±RT): 3858	5-year OS	92.90%	84%	<0.001
Almahariq et al., 2020 [13]	2006–2014	T1–2	BCT: 144,263	5-year OS	94.40%	91.80%	<0.001
		N0	Mastectomy: 87,379	7-year OS	90%	85.20%	<0.001
Current study	2001–2013	T1–2	BCT: 1021	5-year DFS	97%	91%	<0.0005
		N0–1	Mastectomy: 339	10-year DFS	96%	90%	
				5-year OS	97%	95%	0.045
				10-year OS	96%	94%	

* The cohort included up to 5% of patients with N2 disease. Abbreviations: BCT = breast-conserving therapy; BCSS = breast-cancer-specific survival; DFS = disease-free survival; DMFS = distant-metastasis-free survival; OS = overall survival; LRFS = locoregional relapse-free survival; RT = radiation therapy; NA = not available.

## Data Availability

The data that support the findings of this study are not publicly available but can be requested from the corresponding author.

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
