# Peer review of "Improved Survival after Breast-Conserving Therapy Compared with Mastectomy in Stage I-IIA Breast Cancer"

_cancers, 2021, doi:10.3390/cancers13164044_

Round 1

Reviewer 1 Report

The Authors compared effectiveness of breast-conserving therapy (followed by radiotherapy) with mastectomy alone without radiotherapy in the early stages of breast cancer. The results of the study point to the important discovery that even at the very early stage of breast cancer, the primary tumor seems to seed metastatising cancer cells as mastectomy NOT FOLLOWED BY radio- or chemoterapy gives worse results (DFS/OS) although the whole breast is removed together with the tumour nodule(s) and extensive amount of surrounding tissue. It clearly points that some metastatising cancer cells must have already been released in some patients and are able to form the nodules in these patients even if the primary tumour is safely and completely removed. This is a very important issue for breast (and not only) cancer patients and the benefits of after-breast cancer surgery radio-/chemotherapy should be more stressed in the discussion and mentioned in the abstract as I think it is the after-surgery radiotherapy that really makes a difference. What would be the result if mastectomy was also followed by radiotherapy?

Author Response

Reviewer 1: The Authors compared effectiveness of breast-conserving therapy (followed by radiotherapy) with mastectomy alone without radiotherapy in the early stages of breast cancer. The results of the study point to the important discovery that even at the very early stage of breast cancer, the primary tumor seems to seed metastatising cancer cells as mastectomy NOT FOLLOWED BY radio- or chemoterapy gives worse results (DFS/OS) although the whole breast is removed together with the tumour nodule(s) and extensive amount of surrounding tissue. It clearly points that some metastatising cancer cells must have already been released in some patients and are able to form the nodules in these patients even if the primary tumour is safely and completely removed. This is a very important issue for breast (and not only) cancer patients and the benefits of after-breast cancer surgery radio-/chemotherapy should be more stressed in the discussion and mentioned in the abstract as I think it is the after-surgery radiotherapy that really makes a difference. What would be the result if mastectomy was also followed by radiotherapy?

Response 1:

Thank you for pointing out this important aspect. The following sentences were added in a discussion section to stress out the importance of postoperative (adjuvant) treatments even in patients with favourable breast cancer stage/subtypes: “New advances in the systemic treatment, including molecularly targeted therapies, endocrine therapy, taxane-based chemotherapy, and bisphosphonates have all effectively contributed to reduce the risk of distant and local breast cancer recurrence. Consequently, breast cancer mortality rates are declining over the past decades even in patients at low risk of recurrence.”

The importance of post-operative radiation therapy was emphasized in the Discussion section with the following sentence: “In a study by Sun et al., the authors have analysed treatment outcomes of 4,262 patients with clinical stage T1-2N1M0 breast cancer, and 832 (21.6%) of them received mastectomy and postoperative radiation therapy. In multivariate and propensity-score matching analyses, radiation therapy, but not type of surgical treatment, appeared to be an independent prognostic factor for improved OS, DFS and loco-regional recurrence [19].

In addition, the importance of postoperative radiation therapy was mentioned in the Abstract.

Reviewer 2 Report

The authors address a subject quite frequently treated over the past 20 years, the comparison of mastectomy and conservative surgery outcomes in early breast cancer. In particular, they focus on cases with more favourable presentation (stage I-IIA).

The study is retrospective, the two groups are numerically very different and the characteristics of the patients in the two groups are particularly different. These characteristics impose the use of statistical technologies that, in any case, are not able to overcome the difficulties related to the analysed database (no randomisation).

This problem is particularly thoughtful, given the small differences in outcomes, which are related to the good prognosis of the analysed patients. In this regard, some thought could be given to what is statistically significant and what is clinically significant.

I must acknowledge that the authors themselves are aware of all these limitations which they honestly expose in the discussion.

As a specific remark, I would suggest, since it is stressed that the study is performed in the "modern era", to detail more the use of adjuvant chemotherapy and hormone therapy (indications and type). In particular, this information would be useful for chemotherapy, which was apparently used differently in the two groups (Tab 1).

Another suggestion is not to prolong the KM curves too much (almost 15 years in Fig 1 !!!) since the median follow-up is less than 7.5 years.

Author Response

Reviewer 2: The authors address a subject quite frequently treated over the past 20 years, the comparison of mastectomy and conservative surgery outcomes in early breast cancer. In particular, they focus on cases with more favourable presentation (stage I-IIA). The study is retrospective, the two groups are numerically very different and the characteristics of the patients in the two groups are particularly different. These characteristics impose the use of statistical technologies that, in any case, are not able to overcome the difficulties related to the analysed database (no randomisation). This problem is particularly thoughtful, given the small differences in outcomes, which are related to the good prognosis of the analysed patients. In this regard, some thought could be given to what is statistically significant and what is clinically significant. I must acknowledge that the authors themselves are aware of all these limitations which they honestly expose in the discussion. As a specific remark, I would suggest, since it is stressed that the study is performed in the "modern era", to detail more the use of adjuvant chemotherapy and hormone therapy (indications and type). In particular, this information would be useful for chemotherapy, which was apparently used differently in the two groups (Tab 1). Another suggestion is not to prolong the KM curves too much (almost 15 years in Fig 1 !!!) since the median follow-up is less than 7.5 years.

Response 2:

Thank you for your valuable comments to improve the results and discussion sections in our manuscript. We have further expanded the results section, addressing the details type and sequence of chemotherapy and endocrine therapy. Relatively small clinical significance in treatment outcomes (although with statistical significance) was highlighted in the conclusion section. We have modified Figure 1 and adjusted Kaplan-Meier curves as suggested with a shortened time interval on the horizontal (x)-axis.
